# A Systematic Review of Clinical Practice Guidelines for Caries Prevention following the AGREE II Checklist

**DOI:** 10.3390/healthcare11131895

**Published:** 2023-06-30

**Authors:** Demetrio Lamloum, Antonella Arghittu, Pietro Ferrara, Paolo Castiglia, Marco Dettori, Maddalena Gaeta, Anna Odone, Guglielmo Campus

**Affiliations:** 1Department of Restorative, Pediatric and Preventive Dentistry, University of Bern, 3012 Bern, Switzerland; demetrio.lamloum@master.sdabocconi.it (D.L.); madettori@uniss.it (M.D.); 2Department of Public Health, Experimental and Forensic Medicine, University of Pavia, 27100 Pavia, Italy; maddalena.gaeta@unipv.it (M.G.); anna.odone@unipv.it (A.O.); 3Department of Medicine, Surgery and Pharmacy, University of Sassari, 07100 Sassari, Italy; aarghittu@uniss.it (A.A.); paolo.castiglia@uniss.it (P.C.); 4Center for Public Health Research, University of Milan–Bicocca, 20126 Monza, Italy; pietro.ferrara@unipv.it; 5IRCCS Istituto Auxologico Italiano, 20145 Milan, Italy; 6Department of Cariology, Saveetha Dental College and Hospitals, Chennai 600077, India

**Keywords:** public health dentistry, evidence-based dentistry, guidelines, AGREE II

## Abstract

Untreated oral diseases are detrimental to overall well-being and quality of life and are in close relationship with social and economic consequences. The presence of strong evidence for caries primary and secondary prevention is a compulsory tool for the development of clinical practice guidelines (CPGs). This paper was aimed to assess systematically the importance of clinical practice guidelines in caries prevention management considering both the adult and pediatric populations and evaluate them using the Appraisal of Guidelines for Research and Evaluation (AGREE II) Checklist. Records were extracted from EMBASE, SCOPUS, PubMed/Medline and seven other relevant guideline databases between 6 January and 14 February 2023. Two reviewers independently conducted the appraisal using the web-based platform My AGREE PLUS. Twenty-one guidelines/papers met the inclusion criteria and were reviewed. Eight CPGs included both primary and secondary prevention interventions, whereas thirteen presented a single preventive model. Overall, 12 guidelines were published in the USA. The mean AGREE II scores ranged from 35.4% to 84.3%. Of the total twenty-one included guidelines, twelve were classified as “Recommended”, ranging from 56.3% to 84.3%, the others were described as “Recommended with modification”, ranging from 35.4% to 68.9%. From the AGREE II analysis carried out, the CPGs included in this survey adopted a punctual methodological rigor but lacked applicative power. The present survey showed that the public, as the primary beneficiary, played a limited role in the development of the twenty-one CPGs. Hence, methodological improvement can better support high-quality CPG development in the future.

## 1. Introduction

Untreated oral diseases are detrimental to overall well-being and quality of life and are closely associated with social and economic consequences [1,2,3,4]. Recently, the 74th World Health Assembly [5] urged all countries to reorient the traditional treatment-oriented approach towards prevention for timely, comprehensive and inclusive care.

The traditional caries management model focuses on lesion treatment rather than tackling caries as a dysbiosis [6]. Such approach is far outdated due to better understanding of demineralized but structurally intact dentin preservation and the development of microinvasive techniques [6,7,8,9,10]. 

Nonsurgical treatments such as pit and fissure sealants, oral hygiene techniques and fluoride (i.e., silver diamine fluoride) are less dependent on patient behavior [11,12] and are cost-effective [10,13]. 

The presence of strong evidence for primary and secondary caries prevention is a mandatory tool for the development of clinical practice guidelines (CPGs). Defined as “statements containing recommendations to maximize patient care based on a systematic review of the evidence and an assessment of the benefits and harms of alternative treatment options” [14], CPGs are based on the best available scientific evidence. In turn, CPGs must be of proven methodological quality and transparency. Therefore, the international research group Appraisal of Guidelines for Research and Evaluation (AGREE) has introduced a practical tool to develop and assess their quality [15,16,17].

The evaluation of a guideline, in fact, includes judgment about the methods used in its development, the components of the final clinical recommendations and factors related to practical application. [15] 

In fact, the detailed assessment of guideline quality can be complex, and the internal and external validity and applicability can vary depending on several factors (e.g., differences in dental education, outdated concepts, national health policies and reimbursement systems) [7].

The AGREE tool was published in 2003 and amended in 2009 with the goal of developing a tool to support guideline producers so as to limit potential bias in the CPG drafting. This instrument consists of twenty-three items grouped into six quality dimensions. The AGREE tool has been translated into numerous languages, cited in more than 600 publications and endorsed by several health care organizations [15,16,17,18,19].

As a result, recent publications on dental health and caries management guidelines show a considerable interest in comprehensive studies. [20,21]. However, they are limited to European guidelines on fissure sealants and primary prevention in the pediatric population. 

In view of the above, the present paper was designed to evaluate and highlight the importance of clinical practice guidelines in caries prevention management. The AGREE II checklist was used for this purpose. In addition, since most reviews focus only on the pediatric population, the evaluation was also extended to the adult and elderly populations.

## 2. Materials and Methods

### 2.1. Protocol and Registration

The systematic research protocol was registered in the Prospective Register of Systematic Review (PROSPERO ID number 315904). The Preferred Reporting Items for Systematic Reviews and Meta-Analyses (PRISMA) 2020 statement [22] was followed to plan, develop and report the findings of the worldwide available literature on practice guidelines regarding caries prevention in all ages, published from January 2010 to December 2022.

### 2.2. Search Strategy

The search strategy entailed two phases. The first phase involved searching the best available evidence in the literature through the PubMed/Medline, Scopus and Embase databases. The second phase involved searching for guidelines in the relevant guideline databases.

PubMed/Medline, Scopus and Embase were screened using both MeSH terms and free-text keywords included in the search strategy and reported in the Appendix A. The search strategy was adjusted according to the different databases. 

The main guideline databases were searched, namely the National Institute for Health and Clinical Excellence (NICE), Scottish Inter-Collegiate Guidelines (SIGN), Guidelines International Network (GIN), National Clinical Guideline Center (NCGC), Canadian Medical Association Infobase (CMA), Clinical Practice Guidelines Portal (CPGP) and New Zealand Guidelines Group (NZGG), with the following terms: “caries” and “dental health”. Searches were performed from 6 January to 14 February 2023. Cross-reference analysis was performed considering previous systematic reviews about topics close to our research. 

### 2.3. Guidelines Selection and Data Synthesis

A systematic approach was adopted to identify relevant research studies. In particular, the review was conducted in accordance with the methodological process proposed by Arksey and O’Malley [23].

Results were imported into Mendeley reference management software (version 2.89.0), and duplicates were removed. Inclusion and exclusion criteria for the selection of guideline articles were established prior to the literature search. The inclusion criteria were guidelines or articles dealing with the prevention and treatment of caries in cohorts of children, adults and the elderly.

The exclusion criteria were as follows: nonspecific primary research articles focusing on caries; not a full guideline (e.g., review, editorial, guideline summary); guidelines not produced by organizations; other reason (guideline under development; guideline development process or full text not available). 

Each publication was initially assessed for relevance by two members (DL and AA) using the information provided in the abstract. The two reviewers then assessed the relevance of the full text according to the eligibility criteria. Any disagreements were resolved by senior reviewers (GC, MG and PC).

The following key features were extracted from the guidelines: title, country, organization, age target, preventive measures and AGREE II use.

### 2.4. Qualitative Assessment in Adherence to AGREE II Checklist

The quality of practice guidelines that met the inclusion criteria was assessed according to the AGREE II checklist [16,24]. The checklist is the main tool designed to help guideline developers and users to assess the methodological quality of guidelines [25]. The instrument contains six main domains with a total of twenty-three items: (1) scope and purpose, (2) involvement of stakeholders, (3) rigor of development, (4) clarity of presentation, (5) applicability and (6) editorial independence. A complete overview of the checklist and the list of excluded guidelines are available in the Appendix A.

Two reviewers (DL and AA) independently conducted the assessment using the web-based platform My AGREE PLUS (https://www.agreetrust.org/my-agree/, 12 March 2023). Following the AGREE II user manual and instructions [24], each item was scored on a Likert scale from 1 (strongly disagree) to 7 (strongly agree), and the overall average score across all six domains was calculated by the two reviewers for each guideline. Senior investigators (GC, MG and PC) were involved in the assessment until consensus was reached. Inter-rater reliability between the two reviewers was determined using the intraclass correlation coefficient and Cohen’s kappa statistic [26,27]. The articles to be calibrated were randomly selected from the included articles, and a score of 3 was assigned as a threshold. Next, each of the 23 items of the AGREE II checklist was dichotomously assigned a score of 1 or 2. A score of 1 was assigned if the score was equal to or lower than the threshold, and a score of 2 if the score was higher than the threshold. The two calibration parameters ICC (ICC 0.62) and K (Kappa 0.63) were calculated from these results. Results for each of the six domains were given as percentage of the maximum possible score, based on AGREE II user manual. The scores from each domain were determined by combining the scores of each reviewer based on the AGREE II manual formula: (Xob − Xminp)/(Xmaxp − Xminp) × 100, where Xob is “Obtained score”, Xminp is “Minimum score”, and Xmaxp is “Maximum score” [28,29]. As from the literature evidence, a guideline was “Strongly recommended for use in practice” if most domains (four or more) scored above 60%. A guideline was “Recommended for use with some modification” if most domains scored between 30% and 60%. “Not recommended for use in practice” implied that most of the domains of the guideline scored approximately 30% or below [30,31,32].

## 3. Results

### 3.1. Study Selection

The flowchart of the included documents and selection process is presented in Figure 1. 

A total of 1496 reports were retrieved from literature databases, and 19 unique items from guidelines databases. After duplicate removal (224) and titles and abstracts’ screening, seventy-two full texts (sixty-five and seven documents from scientific and guidelines databases, respectively) were assessed for eligibility, and nineteen included; a total of twenty-one documents (two were added after cross-reference) were evaluated [33,34,35,36,37,38,39,40,41,42,43,44,45,46,47,48,49,50,51,52,53].

### 3.2. Study Characteristics

Overall, twelve guidelines were published in the USA [35,37,38,39,40,41,42,43,44,45,46,51], one was Canadian American [52], two from Europe [47,53], two from Japan [34,48], two international [49,50], one from Malaysia [33] and one from Scotland [36]. As shown in Table 1, eight documents presented both modification of individual risk factors for caries development and primary and secondary prevention. Risk factor modification techniques included parents sensibilization about oral health primary prevention, counselling about oral hygiene measures, mainly flossing and interdental brushes, prenatal oral health care for pregnant women and diet [33,36,41,42,47,49,50,52].

Thirteen documents (Table 2) presented a single preventive model. Among these, twelve documents focused on primary prevention and minimally invasive treatments: oral fluoride supplementation, fluoride mouth rinse, fluoride varnish, fluoride gels, fluoride foam, fluoride pastes, caries risk assessment, water fluoridation, glass ionomer cement (GIC), pit-and-fissure sealants, minimally invasive restorative treatments and silver diamine fluoride (SDF) [34,35,37,38,39,40,43,45,46,48,51,53]. 

Recommendations about the use of indirect pulp cap, direct pulp cup and medicaments for treating caries development were reported by one guideline [44]. 

### 3.3. Results of Syntheses 

The mean AGREE II scores ranged from 35.4% to 84.3%. Of the total twenty-one included guidelines, twelve were classified as “Recommended”, ranging from 56.3% to 84.3%, whereas, as shown in Table 3, others were described as “Recommended with modification”, ranging from 35.4% to 68.9%.

#### 3.3.1. Domain 1—Scope and Purpose

Overall, the mean score was 85%. Two guidelines reached 100% for the description of the objectives, health question and target population [38,51]. Eleven out of the twelve guidelines classified as “Recommended” reached a score above 60% [33,35,36,43,44,46,47,48,49,50,52], whereas all guidelines classified as “Recommended with modifications” exceeded 70%.

#### 3.3.2. Domain 2—Stakeholder Involvement

For this domain, the total average score was 63.5%. Target users of the guideline were clearly defined in all except for three publications [35,37,48]. Guideline development groups were heterogeneous, including individuals from all relevant professional groups, such as for example epidemiologists, dental hygienists, oral public health experts and public health analysists. Poorer scores were mainly attributable to the scarce involvement of the views and preferences of the target population. “Recommended” guidelines scored on average between 60.5% and 91.1%, while “Recommended with modifications” scored between 16.7% and 94.4%.

#### 3.3.3. Domain 3—Rigor of Development

Overall, Rigor of Development scored between 29.1% and 89%. The vast majority of them had a clear statement on the literature searching strategies, including databases and eligibility criteria for full-text analysis [33,34,35,36,37,39,40,45,46,48,49,50]. Information on the external reviewing process was reported in seven guidelines [33,34,36,44,46,48,52], with the inclusion of the reviewers and their affiliation. Lastly, a complete revision and updating process of the guidelines with explicit timeline criteria and methods was presented in five documents [34,44,45,46,48]. Only one “Recommended” guideline scored poorly [52], whilst two among those “Recommended with modifications” exceeded 60% [34,45].

#### 3.3.4. Domain 4—Clarity of Presentation

Clarity of Presentation reached on average 84.9%. All the guidelines appeared specific and unambiguous. The different options for management of the condition or health issue were clearly presented, and a concrete and precise description of which option was appropriate in which situation and in what population group was also presented. Key recommendations were easily identifiable. Only one record scored below 60% [46]. 

#### 3.3.5. Domain 5—Applicability

Applicability was evaluated with a mean score of 45.5%. Information on opportunities and barriers for the application were reported in two documents [38,39], and about potential resource implications of applying the recommendations in four documents [37,39,44,46]. Advice and/or tools on how the recommendations can be put into practice and the guideline monitoring and/or auditing criteria were reported in five documents [35,37,40,44,46].

Except for one record [36], the retrieved guidelines did not provide advice or tools on how the recommendations can be put into practice in a dedicated implementation section. Few considered potential resource implications of applying the recommendations, likewise for monitoring and auditing criteria. All of the “Recommended with modifications” guidelines scored below fifty percent, while seven out of twelve of the “Recommended” guidelines scored above fifty percent [33,36,37,43,44,46,48].

#### 3.3.6. Domain 6—Editorial Independence

The average Editorial Independence was 54.2%. Clear statements about the source of fundings for guidelines process were reported in nine records [33,38,44,46,48,50,51,52,53]. General statements about conflicts of interest were normally considered; however, individual group members did not declare whether they had any competing interests between them. Two out of twelve in the “Recommended” group scored 25.0% or below [43,46], whereas all “Recommended with modifications” guidelines scored 50.0% or below.

## 4. Discussion

Clinical guidelines represent the actualization into practice of the outcomes derived from clinical research. Starting from this premise, several scientific societies prepared, published and amended *ad hoc* specific guidelines. Clinical guidelines are usually developed via specific methods (e.g., consensus procedures, Delphi method, etc.). Since clinical guidelines should set the standard care for the individual patient, the community and health care providers, they must follow a strict methodological and systematic approach. Thus, the AGREE II is now the international tool for the assessment of practice guidelines.

In this paper, the AGREE II method was applied to twenty-one guidelines on caries prevention and treatment.

None of the CPGs were found to adequately address all six of the domains when using the AGREE II appraisal instrument. The overall quality of the recommendations was average (mean 65.7%). Twelve of the twenty-one selected guidelines were “Recommended” and nine “Recommended with modifications”. 

Domain 1 (Scope and Purpose) was clearly described in all guidelines included in the survey. In some cases, the target population of the guidelines included several population cohorts, such as children and adolescents [33,35,36,38,40,41,43,44,45,46,47,50,51,52,53], and in three cases, only the adult cohort was included [34,48,49]. Other CPGs were nonspecific in their description of the target group, extending the recommendation to a wider audience [37,39]. CPGs should analyze aspects related to patients’ perspectives, including their experiences and expectations (e.g., collected through questionnaires). However, despite the fact that the description of the items in Domain 1 does not strictly follow the grid proposed by the AGREE II method, the twenty-one guidelines included performed excellently. The authors correctly aligned the objectives of each guideline with the key recommendations.

With respect to Stakeholder Involvement, some variability was observed in the definition of the roles of recipients and facilitators. Guideline (CPG) documents that address public health aspects (e.g., water fluoridation) emphasize the involvement of the recipients. The expert panel involved in the USA Public Health Service Guideline for Fluoride Concentration in Drinking Water for the Prevention of Dental Caries clearly and comprehensively interpreted the opinions of the public. The public’s knowledge and opinions were gathered from 19,300 responses collected electronically through an online survey [39]. The health risk perceived by the public was considered in the development of the recommendations for the proposed standards for drinking water fluoridation. Public involvement in the development and drafting of guidelines, although complex to implement, is of fundamental importance. Publicly available CPGs can be a useful tool for improving patient compliance and increasing the magnitude of the economic and health benefits of recommended preventive and/or therapeutic practices [54,55].

The CPGs followed systematic methodological criteria for the collection of data from multiple databases and by multiple reviewers [33,34,35,36,37,44,45,48]. The selection of external reviewers and the definition of adverse effects and risks were rarely considered, which reduced the overall domain score. 

The involvement of external clinical experts and methodologists increases the accuracy of the content of CPGs [56]. Only one document mentioned the use of external reviewers [34]. Details of comments and corrections were not reported. The variables related to the benefits of caries prevention and treatment were reported. However, side effects were rarely explicitly mentioned. The most relevant was fluorosis [35,39,44,46]. When considering the quality of the results, special attention should be paid to the rigor of development [57]. A high score for this domain indicates minimal bias and evidence-based design during the development process, linking recommendations and supporting evidence. Therefore, few CPGs [38,39,40,41,42,51,53] received low scores for this domain because the information was not clearly stated. Implementation of CPGs cannot be separated from consideration of the benefits, side effects and risks implicit in the recommendations (e.g., primary outcome, quality of life, adverse effects of treatments, symptom management and/or discussion of different models of care). Although it may seem more practical to emphasize preventive measures, a clear description of responsibilities is necessary [58]. Clarity of presentation received a high average score. The content of CPGs was described in an explicit and coherent way, considering tables, specific sections and paragraphs, thus improving reproducibility. This aspect is crucial to easily identify most relevant recommendations [58].

In terms of applicability, few CPGs included an economic evaluation of the direct costs of preventive interventions [37]. Economic analyses are rarely considered in preventive dentistry. Further economic evaluations are needed and should be included in CPGs [59]. 

Guidelines dissemination and implementation are not examined. Standardized guidelines implementation methods are not available. However, those instruments should be weighted for readers’ use (e.g., clinicians, patients) [60]. In addition, many items in the Applicability domain refer to facilitators and the description of possible obstacles regarding the application of the CPGs.

Editorial Independence may be a difficult topic to handle, especially if the body that publishes the CPG is the same one that promotes its contents. Although financial interests are often the most obvious, intellectual interests are increasingly recognized and may be powerful motivators for researchers, systematic reviewers and guideline authors [61]. Support may pertain to the entire project or only to certain aspects.

It should be explicitly stated that the opinions and material interests of the funder did not influence the content of the final recommendations. In addition, several items in this domain specifically require a description of the types of conflicting interests that were considered, the methods by which potential conflicting interests were sought, a description of the conflicting interests, and a description of how these might have influenced the guideline and recommendation development process. 

Although all CPGs stated that they were free of conflicts, none of the CPGs provided any readily identifiable elements that could be traced back to the answers to these questions. Few CPGs explicitly stated that the funding body had not influenced the content of the guideline [33,38,44,46,48,50,51,52,53]. The final grading of CPGs using the AGREE II checklist as “Recommended” or “Recommended with modification” does not directly imply a better or worse quality of evidence; it means that the authors did not strictly follow or adhere to AGREE II during the development process, and it encourages improved reporting for subsequent updates. Nevertheless, the scientific objectivity of the recommendations included in the CPG is the best tool to avoid any suspicion of conditioning [14,60].

### Study Limits

During the review process, the study group identified some strengths and weaknesses. First, the AGREE II checklist proved to be an extremely clear and user-friendly tool, thus facilitating CPG qualitative assessment. The clarity of the items and appraisal yardsticks improved reviewers’ judgement uniformity. Secondly, the inclusion of guideline-specific databases supported the retrieval of CPGs. Furthermore, to the reviewers’ knowledge, the inclusion of the adult population beside the pediatric one is innovative and was never investigated before. 

The main weakness is the reviewer-intrinsic variability in the application of the AGREE II checklist. Although this method is a useful tool in the analysis of the CPGs by means of a detailed list of items to which the reviewer assigns a score, the result of the overall scores could be influenced by a subjective evaluation criterion. In fact, the CPGs were reviewed by two evaluators, and discrepancies emerged with respect to some items, which persisted even after the second round of discussion and review by external observers. In addition, AGREE II has no standardized method to assess the strength of a recommendation; the authors decided to use previous published methods [30,32]. This procedure can be postulated by other researchers. Thus, some elements could be sensitive to personal interpretation and may not be evaluated uniformly by all reviewers; the decision was taken to award the scores as unanimously as possible and to analyze all CPGs included in this research by applying the same evaluation criterion. To reduce possible differences even more, a priori calibration of reviews was performed. In some CPGs, supplementary materials were easily obtainable.

## 5. Conclusions

This manuscript confirms the main recommendations in the areas of oral health promotion and primary and secondary prevention of caries. Factors such as a low-carbohydrate diet and oral hygiene, especially prenatal and in childhood, and parental figures are of paramount importance with respect to a child’s oral health. As highlighted in this review, the use of topical or systemic fluoride products has an extensive scientific basis in the literature. Moreover, in the adult population, advances in clinical cariology have allowed clinicians to diagnose caries at an earlier stage, thus enhancing the role of CPGs in the application of minimal invasive techniques.

From the AGREE II analysis carried out, the CPGs included in the survey adopted a punctual methodological rigor but lacked applicative power. From the present survey, the public, as the primary beneficiary, played a limited role in the development of the twenty-one CPGs. Furthermore, CPG implementation process is not specified, except in a few records. It may be useful to involve the public and/or the general population in the drafting of caries prevention CPGs also in the community context. In fact, since the population in this case represents both the target audience and stakeholders, involving patients, by making them active and diligent participants in the recommendation development, could not only ensure the completeness of an AGREE II guideline but also represent a strategic tool for training and compliance. Our research highlighted the need to create in the near future a global “consensus” from the point of view of caries treatment based on variables such as the prevalence of the disease in different geographical areas but also for health education and health literacy.

## Figures and Tables

**Figure 1 healthcare-11-01895-f001:**
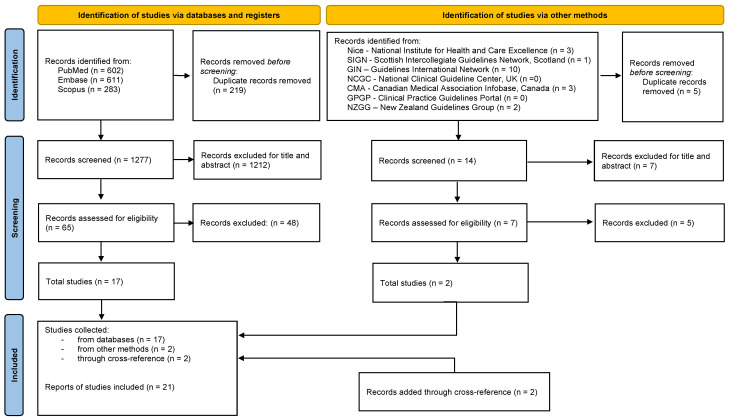
Flow diagram of PRISMA 2020 [22].

**Table 1 healthcare-11-01895-t001:** List of guidelines and consensus papers: mixed models.

Title	Country	Organization *	Age Target	Preventive Measures ^#^	AGREE II Use
How to intervene in the caries process in children: A joint ORCA and EFCD expert Delphi consensus statement [50]	International	EFCD & ORCA	0–12 yy	(A) Parents sensibilization about oral health primary prevention, fluoride toothpaste for children (B) Fluoride varnish, silver diamine fluoride, sealants, composite strip crowns, GIC, nonrestorative caries control(C) Composites	NO
How to intervene in the caries process in adults: proximal and secondary caries? An EFCD-ORCA-DGZ expert Delphi consensus statement [49]	International	EFCD & ORCA	Adult	(A) Oral hygiene measures, mainly flossing and interdental brushes, diet (B) Fluoride supplementation, sealants, caries sealing, caries infiltration (C) Restorative treatments (using resins and amalgam) and indirect restorations	NO
Early childhood caries in indigenous communities [52]	USA and Canada	AAP	0–6 yy	(A) Prenatal oral health care for pregnant women, oral health education (B) Water fluoridation, fluoride varnish, sealants, interim therapeutic restorations, silver diamond fluoride	NO
Guidelines on the use of fluoride for caries prevention in children: an updated EAPD policy document [47]	Europe	EAPD	0–18 yy	(A) Fluoride toothpaste (B) Fluoride supplementation (Fluoride gels, rinses and varnishes, water fluoridation fluoridated milk, fluoridated salt, fluoride tablets/lozenges and drops)	NO
Guideline on restorative dentistry [42]	USA	AAPD	0–18 yy	(A) Caries Risk Assessments (CAF) (B) Sealants, PMC, GIC (C) Resin infiltration, amalgam, composites	NO
Dental interventions to prevent caries in children [36]	United Kingdom (Scotland)	HIS or NICE	0–18 yy	(A) Oral health promotion (B) Caries Risk Assessment, sealants, fluoride varnish, chlorhexidine varnish, low-release fluoride beads, fluoride gel, fluoride drops or tablets, fluoride mouthwash	NO
Management of severe early childhood caries [33]	Malaysia	MH	0–6 yy	(A) Diet, good oral hygiene, use of fluoridate toothpaste (B) CRA, GIC, SSC, check-ups	YES
Guideline on fluoride therapy [41]	USA	AAPD	0—18 yy	(A) Fluoride toothpaste (B) Fluoride dietary supplements, fluoride gel, fluoride mouthrinse	NO
Evidence-based consensus for treating incipient enamel caries in adults by non-invasive methods: recommendations by GRADE guideline [48]	Japan	JSCD	Adult	(B) Topical fluoride, fluoride GIC, sealants	YES

^#^ Preventive measures: (A) risk factor modification; (B) primary prevention and minimally invasive treatments; (C) invasive techniques.* Organization: AAPD—American Academy of Pediatric Dentistry; HIS or NICE—Healthcare Improvement Scotland; MH—Minister of Health; EAPD—European Association Pediatric Dentistry; EFCD—European Federation of Conservative Dentistry; ORCA—Organization of Caries Research; JSCD—Japanese Society of Conservative Dentistry; AAP—7 American Academy of Pediatrics.

**Table 2 healthcare-11-01895-t002:** List of guidelines and recommendations presenting a single preventive model: noninvasive, microinvasive techniques and invasive techniques.

**Title**	**Country**	**Organization ***	**Target**	**Preventive Measures ^#^**	**AGREE II Use**
U.S. public health service recommendation for fluoride concentration in drinking water for the prevention of dental caries [39]	USA	USDH	0–18 yy/Adult	(B) Water fluoridation	NO
Evidence-based clinical practice guideline for the use of pit-and-fissure sealants: A report of the American Dental Association and the American Academy of Pediatric Dentistry [43]	USA	ADA	0–18 yy	(B) Pit-and-fissure sealants	YES
Prevention of dental caries in children from birth through age 5 years: US Preventive Services Task Force recommendation statement [38]	USA	USPSTF	0–5 yy	(B) Oral fluoride supplementation, fluoride varnish	NO
Use of Silver Diamine Fluoride for Dental Caries Management in Children and Adolescents, Including Those with Special Health Care Needs [46]	USA	AAPD	0–18 yy	(B) Silver Diamine Fluoride	YES
Evidence-based clinical practice guideline on nonrestorative treatments for carious lesions: A report from the American Dental Association [37]	USA	ADA	0–18 yy Adult	(B) Sealants	YES
Topical fluoride for caries prevention: executive summary of the updated clinical recommendations and supporting systematic review [35]	USA	ADA	0–18 yy	(B) Fluoride mouth rinse, fluoride varnish, fluoride gels, fluoride foams and fluoride pastes	NO
Guideline on Caries-risk Assessment and Management for Infants, Children, and Adolescents [40]	USA.	AAPD	0–18 yy	(B) Caries Risk Assessment	NO
Screening and interventions to prevent dental caries in children younger than 5 years: US preventive services task force recommendation statement [51]	USA	USPSTF	0–5 yy	(B) Oral fluoride supplementation, fluoride varnish	NO
Best clinical practice guidance for clinicians dealing with children presenting with molar-incisor-hypomineralisation (MIH): an updated European Academy of Pediatric Dentistry policy document [53]	Europe	EAP D	0–18 yy	(B) Sealants, GIC, PMC, resine restorations (composites)	NO
Use of Pit-and-Fissure Sealants [45]	USA	AAPD	0–18 yy	(B) Sealants	YES
Clinical guidelines for treating caries in adults following a minimal intervention policy--evidence and consensus-based report [34]	Japan	JSCD	Adult	(B) Caries treatment with Minimally Invasive Policy	NO
Use of Vital Pulp Therapies in Primary Teeth with Deep Caries Lesions [44]	USA	AAPD	0–18 yy	(C) Indirect Pulp Cap, direct pulp cup, medicaments	YES

^#^ Preventive measures: (A) risk factor modification; (B) primary prevention and minimally invasive treatments; (C) invasive techniques. * Organization: AAPD—American Academy of Pediatric Dentistry; ADA—American Dental Association; EAPD—European Association Pediatric Dentistry; USDH—USA Department of Health and Human Services Federal Panel on Community Water Fluoridation; JSCD—Japanese Society of Conservative Dentistry; AAP—American Academy of Pediatrics; USPSTF—US Preventive Services Task Force.

**Table 3 healthcare-11-01895-t003:** Distribution of guidelines for caries prevention and treatment by AGREE II checklist scores (scores are expressed in %).

Organization/Year/References of Guidelines	D.1	D.2	D.3	D.4	D.5	D.6	Mean (SD)	Overall Evaluation
American Dental Association. 2018 [37]	55.5	91.1	73.7	99.4	89.4	96.6	84.3 (16.7)	Recommended
Healthcare Improvement Scotland (HIS) or NICE. 2014 [36]	88.9	88.9	70.8	88.8	95.8	66.7	83.3 (11.7)	Recommended
American Academy of Pediatric Dentistry. 2017 [44]	75.6	65.3	88.0	87.0	90.0	96.0	83.7 (11.2)	Recommended
European Academy of Paediatric Dentistry. 2019 [47]	92.2	64.7	68.6	99.9	49.5	99.9	79.1 (21.1)	Recommended
EFCD- ORCA. 2020 [49]	88.7	66.6	64.3	99.9	34.8	99.8	75.7 (25.4)	Recommended
EFCD- ORCA. 2020 [50]	88.8	57.5	67.8	99.9	33.3	99.8	74.5 (26.5)	Recommended
The Japanese Society of Conservative Dentistry. 2020 [48]	62.7	66.7	89.0	68.9	59.6	99.1	74.3 (15.9)	Recommended
American Dental Association. 2013 [35]	61.8	86.7	80.3	78.3	34.6	99.8	73.6 (22.7)	Recommended
Malaysia Ministry of Health. 2012 [33]	88.9	61.1	75.0	77.8	70.8	58.3	72.0 (11.3)	Recommended
American Academy of Pediatrics. 2021 [52]	99.9	63.3	29.8	96.6	45.8	88.2	70.6 (28.9)	Recommended
American Dental Association. 2016 [43]	88.8	77.8	62.5	88.8	66.7	25.0	68.3 (23.8)	Recommended
American Academy of Pediatric Dentistry. 2017 [46]	87.2	60.5	67.9	58.3	63.7	0.0	56.3 (29.4)	Recommended
US Preventive Services Task Force. 2021 [51]	100.0	94.4	43.7	100.0	25.0	50.0	68.9 (33.2)	Recommended with modifications
US Preventive Services Task Force. 2014 [38]	100.0	77.8	35.4	72.2	50.0	50.0	64.2 (23.5)	Recommended with modifications
The Japanese Society of Conservative Dentistry. 2012 [34]	94.4	55.5	79.2	83.3	41.6	8.3	60.4 (32)	Recommended with modifications
U.S. Department of Health and Human Services Federal Panel 2015 [39]	94.4	77.8	47.9	72.2	58.3	16.7	61.2 (27.1)	Recommended with modifications
European Academy of Paediatric Dentistry. 2019 [53]	88.9	61.1	56.2	61.1	0.0	50.0	52.9 (40.7)	Recommended with modifications
American Academy of Pediatric Dentristry. 2018 [45]	88,9	33,3	83.3	94.4	8,3	8.3	52.8 (37.3)	Recommended with modifications
American Academy of Pediatric Dentistry. 2016a [40]	88.8	33.3	43.9	99.9	37.5	0.0	50.6 (3.2)	Recommended with modifications
American Academy of Paediatric Dentistry. 2016c [42]	77.8	33.3	33.3	77.8	0.0	8.3	38.4 (33.2)	Recommended with modifications
American Academy of Pediatric Dentristry. 2016b [41]	72.2	16.7	29.1	77.8	0.0	16.7	35.4 (32.1)	Recommended with modifications
Mean (including Recommended and Recommended with modifications)	85.0	63.5	61.4	84.9	45.5	54.2	65.7 (16.1)	-

Domain: D.1–6. D.1: Scope and Purpose; D.2: Stakeholder involvement; D.3: Rigor of Development; D.4: Clarity of Presentation; D.5: Applicability; D.6: Editorial Independence.

## Data Availability

All the data of the paper are available on the Appendix A.

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
