# Peer review of "A Systematic Review of Clinical Practice Guidelines for Caries Prevention following the AGREE II Checklist"

_healthcare, 2023, doi:10.3390/healthcare11131895_

Round 1
Reviewer 1 Report
Manuscript title: A systematic review of clinical practice guidelines for caries prevention following the AGREE II Checklist
This systematic review is a novel approach for checking and evaluating guidelines for caries prevention. However, there are few points to be considered,
Abstract: Well-written and within the word limits. However, there is a bracket in line 31 that does not have any significance. Keywords: simple and understandable.
Introduction: In this section authors have explained the need for the study clearly. However, they have not mentioned the importance of AGREE II checklist and how it will help in evaluation and prevention of primary and secondary caries. Authors should add more references to support their study.
Material and method: This section is well written and explained.
Results: Well written and every point is explained.
Discussion: This section fails to support the finding of the study and needs to be improved.
Conclusion: Well-written and supports the findings of the review.
References are correctly marked, and no duplication is seen.
This article is poorly written and requires English editing. many sentences in introduction, results and discussion does not make sense.
Reviewer 2 Report
1: Some phrases have grammatical errors or not easy to read ( highlighted in the pdf)
2: table 2 is not clear (using the numbers before the preventive measure is a bit confusing) and the * is aimed at all abbreviations not only the AGREE use?

Some phrases have grammatical errors or not easy to read ( highlighted in the pdf)
Reviewer 3 Report
The authors have performed a systematic review of CPG for caries prevention of all ages. They included 21 such guidelines and asks for some methodological improvement in the development of such CPGs in the future.
This paper is interesting and of good quality. It is registered in Prospero and follows the PRISMA-standard.
I think the way you use AGREE II is in line with practice, but not always according to the manual and that this should be mentioned.
- - AGREE II recommends that preferable 4 appraisers check each guideline (even though 2 is acceptable.)
- - The manual states that “The six domain scores are independent and should not be aggregated into a single quality score”. But that is what you do when you present mean AGREE II scores.
- - I think the cut-off values for “recommendation etc” is not according to the AGREE II manual, where this rating should not be mathematically calculated (for example above 60%) but assessed with care (some domains may be more important than others for example).
Some minor language errors at the end. What supplementary materials can be downloaded (not written out). Funding: remove “Please add”
Reviewer 4 Report
This review targets a very important topic, caries prevention management. Moreover, it is clearly written.
I have few comments to add:
· There is no need to remind the study aim at the beginning of the discussion section. Please present there the main findings, which reply to the study aim.
· Please discuss the weak points and limitations the study, mainly the biases of systematic reviews.
